# Inbred Selection for Increased Resistance to Kernel Contamination with Fumonisins

**DOI:** 10.3390/toxins15070444

**Published:** 2023-07-04

**Authors:** Rogelio Santiago, Antonio J. Ramos, Ana Cao, Rosa Ana Malvar, Ana Butrón

**Affiliations:** 1Misión Biológica de Galicia (CSIC), Carballeira 8, Salcedo, 36143 Pontevedra, Spain; rsantiago@mbg.csic.es (R.S.); anacao@mbg.csic.es (A.C.); rmalvar@mbg.csic.es (R.A.M.); 2Applied Mycology Unit, Department of Food Technology, Engineering and Science, University of Lleida, Agrotecnio-CERCA Center, Av. Rovira Roure 191, 25198 Lleida, Spain; antonio.ramos@udl.cat

**Keywords:** Fusarium ear rot, fumonisin, pedigree selection, hybrid performance

## Abstract

In temperate world-wide regions, maize kernels are often infected with the fumonisin-producing fungus *Fusarium verticillioides* which poses food and feed threats to animals and humans. As maize breeding has been revealed as one of the main tools with which to reduce kernel contamination with fumonisins, a pedigree selection program for increased resistance to Fusarium ear rot (FER), a trait highly correlated with kernel fumonisin content, was initiated in 2014 with the aim of obtaining inbred lines (named EPFUM) with resistance to kernel contamination with fumonisins and adapted to our environmental conditions. The new released EPFUM inbreds, their parental inbreds, hybrids involving crosses of one or two EPFUM inbreds, as well as commercial hybrids were evaluated in the current study. The objectives were (i) to assess if inbreds released by that breeding program were significantly more resistant than their parental inbreds and (ii) to examine if hybrids derived from EPFUM inbreds could be competitive based on grain yield and resistance to FER and fumonisin contamination. Second-cycle inbreds obtained through this pedigree selection program did not significantly improve the levels of resistance to fumonisin contamination of their parental inbreds; however, most EPFUM hybrids showed significantly better resistance to FER and fumonisin contamination than commercial hybrids did. Although European flint materials seem to be the most promising reservoirs of alleles with favorable additive and/or dominance effects for resistance to kernel contamination with fumonisins, marketable new Reid × Lancaster hybrids have been detected as they combine high resistance and yields comparable to those exhibited by commercial hybrids. Moreover, the white kernel hybrid EPFUM-4 × EP116 exploits the genetic variability within the European flint germplasm and can be an alternative to dent hybrid cultivation because white flint grain can lead to higher market prices.

## 1. Introduction

In temperate world-wide regions, maize kernels are often infected with the fumonisin-producing fungus *Fusarium verticillioides,* where the accumulation of fumonisins in the kernel above safety levels poses food and feed threats to animals and humans [1,2,3,4,5]. Several strategies can be managed to control kernel contamination with fumonisins and, among them, breeding has emerged as one of the most effective [6,7,8,9]. High kernel contamination with fumonisins is strongly associated with increased Fusarium ear rot (FER) disease caused by the fungus, as high genotypic correlation coefficients between the visual score for FER and fumonisin content have been reported across environments and genetic backgrounds [10,11,12,13]. Consequently, as fumonisin quantification is expensive and labor-consuming, FER is commonly used as the target trait in breeding programs to indirectly reduce fumonisin contamination.

Previous studies have reported that additive, dominance and epistatic effects are involved in maize resistance to FER [14,15,16,17]. Nevertheless, as additive effects are, in general, more important and stable than dominance effects are, and moderate to high genotypic correlation coefficients have been reported between inbred lines and testcrosses, breeding programs for increased resistance to FER have mainly focused on additive effects [11,18,19]. Introgressions from a resistant donor of resistance to FER (GE440) to a commercial inbred (FR1064) resulted in some inbred lines with improved disease resistance [20]. Similarly, a pedigree selection approach to breed inbred lines with increased resistance to FER and fumonisin accumulation resulted in maize genotypes more resistant to fumonisin accumulation in some environments [21]. Phenotypic recurrent selection for reduced FER and increased yield in the ReFus synthetic population, partially resistant to FER, attained significant gains for resistance to FER and indirectly for resistance to fumonisin accumulation in the population per se and in testcrosses [22]. Posteriorly, Butoto et al. [23] showed that genomic and phenotypic selection schemes applied to the ReFus population were similar and highly effective at reducing FER and fumonisin content. Based on the few reports on the effectiveness of breeding programs for increased resistance to FER, breeding efforts to reduce kernel contamination with fumonisins have been scarce and new initiatives are necessary to address farmers’ needs. Therefore, a pedigree selection program for increased resistance to FER was initiated by the breeding group of Misión Biológica de Galicia-CSIC in 2014 with the aim of obtaining inbred lines with resistance to kernel contamination with fumonisins. Because heritabilities for FER can be considerably reduced under natural conditions compared to artificial inoculation [12,24], selection was carried out under artificial inoculation conditions in order to achieve the highest efficacy. The main objectives of the current study were (i) to assess if inbreds released by that breeding program were significantly more resistant than their parental inbreds were and (ii) to examine if hybrids derived from EPFUM inbreds could be competitive based on grain yield and resistance to FER and kernel fumonisin contamination.

## 2. Results

In general, second-cycle inbred lines tended to improve resistance to FER and fumonisin contamination compared to their parental inbreds, although differences were not significant among either groups (defined in Table 1) or inbreds within groups (Figure 1 and Figure 2). The interactions of year × group and year × inbred (group) were also not significant.

Contrarily to inbreds, hybrids significantly differed for all traits, except for FER (Table 2). The interaction of year × hybrid was significant for days to silking and FER. Hybrids involving experimental inbreds presented lower fumonisin contents than commercial hybrids did, except EPFUM-3 × EP116 (Figure 3). Specifically, hybrids EPFUM-3 × EPFUM-8, EPFUM-3 × EPFUM-10, EPFUM-7 × EPFUM-1, EPFUM-7 × EPFUM-9, EPFUM-8 × EPFUM-1, EPFUM-8 × EPFUM-9, EPFUM-9 × EPFUM-2, EPFUM-10 × EPFUM-1, EPFUM-11 × EPFUM-1 and EPFUM-12 × EPFUM-2 were significantly less contaminated than was any commercial hybrid. Eight of these hybrids were crosses involving EPFUM inbreds developed from European flint materials (EPFUM-1 to 4). In addition, five hybrids involving experimental inbreds did not significantly differ for grain yield from the most productive commercial hybrid (Figure 4). These included EPFUM-6 × EPFUM-10, EPFUM-7 × EPFUM-9, and EPFUM-8 × EPFUM-9 corresponding to the Reid × Lancaster pattern, EPFUM-4 × EPFUM-9 to the European flint × Reid pattern, and the white-kernel hybrid EPFUM-4 × EP116 being a cross between two European flint inbreds. Hybrids EPFUM-8 x EPFUM-9 and EPFUM-7 × EPFUM-9 stood out for being among the best hybrids for grain yield and resistance to kernel fumonisin contamination. In addition, these hybrids presented good adaptation to cultivation in Northwestern Spain as days to shedding pollen ranged between 72 and 75 days and average kernel moisture ranged between 17.3 and 19.4 (Table 3).

The Pearson correlation coefficient between fumonisin content and FER was high among inbreds (N = 24, r^2^ = 0.76, *p* < 0.0001) and moderate among hybrids (Table 3). Pearson correlation coefficients between hybrid performance and mid-parent and heterosis estimates for FER and fumonisin content (Table 4) were computed with data collected on the 22 hybrids involving crosses between EPFUM inbreds (Table 5). High correlations were observed between heterosis and hybrid performance for FER and fumonisin content; meanwhile, correlation coefficients between hybrid performance and mid-parent estimates were low and not significant (Table 6).

## 3. Discussion

In the current study, second-cycle inbreds obtained through a pedigree selection program did not significantly improve the resistance to fumonisin contamination of their parental inbreds, but those parents were already the most resistant to fumonisin contamination among a diverse collection of inbred lines with more than 200 accessions [24]. This result did not seem to be a consequence of performing indirect selection using FER as the target trait because the correlation coefficient between FER and fumonisin content in the inbreds was high according to the previous literature [24,25,26]. More likely, selection based on single plant performance could be responsible for the weak resistance changes observed in the EPFUM inbreds compared to that of their parentals because low heritability estimates on an individual plant basis were reported for FER [26]. In this sense, an inbred selection scheme based on replicated family evaluation for FER resulted in a trend toward improved resistance to fumonisin accumulation, but with limited effectiveness [20].

On the other hand, most hybrids involving crosses among EPFUM inbreds showed better resistance to FER and fumonisin contamination than did commercial hybrids. Butoto et al. [24] suggested that breeders can effectively select inbreds for FER and FUM resistance, reserving the development and evaluation of topcross hybrids at the later stages of the breeding program. However, among the 22 hybrids developed from EPFUM inbreds, the correlation coefficients between the mid-parent value and hybrid performance for FER and fumonisin content were not significant; meanwhile, hybrid performance was highly correlated with mid-parent heterosis, suggesting that differences among these hybrids were driven predominantly by dominance and/or epistatic effects.

Among the best 10 hybrids for fumonisin content, eight were crosses between European flint inbreds and Lancaster or Reid inbreds, indicating that European flint materials are reservoirs of alleles with favorable additive and/or dominance effects. In agreement, Santiago et al. [25] studied the performance of a broad collection of inbred lines under inoculation with *F. verticillioides* and observed that flint inbreds had significantly (*p* < 0.10) less fumonisin content and FER than did the dent genotypes.

However, although hybrids with higher resistance to fumonisin contamination are highly desirable, resistance has to be accompanied by high grain yields to make hybrids commercially competitive. We could check that all commercial hybrids tested, with the exception of “Oldham”, showed higher yields but also higher fumonisin contamination. Within this context, Reid × Lancaster hybrids EPFUM-8 × EPFUM-9 and EPFUM-7 × EPFUM-9 were the most marketable hybrids because they combined high resistance and yields comparable to those exhibited by commercial hybrids. Hybrids EPFUM-6 × EPFUM-10, EPFUM-4 × EPFUM-9 and EPFUM-4 × EP116 could also be commercially interesting because these hybrids displayed kernel fumonisin contents that were significantly lower than those shown by the most productive commercial hybrid, “DA SPICIO”, but displayed comparable grain yields. EPFUM-4 × EP116 is a good example of heterosis within the European flint group as EPFUM-4 was derived from the cross of the inbred F575, developed from the French landrace “Millete Lauragais”, and the inbred EP65, developed from the Portuguese landrace “Regional de Fafe”; meanwhile, EP116 comes from a northern Spanish landrace, “Enano norteño × Vasco” [27]. The genetic variability within the European flint germplasm between EP116 and EPFUM-4 and their good adaptation to northwestern Spain, where these inbreds have been bred, have been capitalized on with the hybrid EPFUM-4 × EP116. Hybrid EPFUM-4 × EP116 can be an alternative higher-value specialty crop and add genetic variability to the current narrow-based commercial dent hybrids being grown [28]. The EPFUM-4 × EP116 hybrid would follow the western Europe × northern Spain pattern that in a previous study was represented by the landrace cross “Lazcano × Millette Lauragais” which was the most promising landrace cross across sites in that study [29]. In addition, another advantage of the hybrid EPFUM-4 × EP116 is its white kernel color, because flint white varieties are preferred for human consumption purposes in many places of the world [30,31]. In Europe, this hybrid could be particularly interesting in areas where the human consumption of maize as bread or polenta is common to reduce the associated risks of fumonisin contamination [32,33].

## 4. Conclusions

Second-cycle inbred lines from the three heterotic groups tested showed a non-significant improvement in resistance to FER and fumonisin contamination compared to their parental inbreds, probably due to the low inheritance of maize resistance to FER. Therefore, breeding schemes using replicated family evaluations for FER or genomic selection are the new approaches to be explored. However, hybrids involving crosses among the second-cycle lines showed lower contamination with fumonisins in comparison to commercial hybrids. This indicates the importance of dominance and epistatic factors for improving resistance to fumonisin contamination. European flint materials seem to be the most promising reservoirs of alleles with favorable additive and/or dominance for resistance to kernel contamination with fumonisins. The white kernel hybrid EPFUM-4 × EP116 exploits the genetic variability within the flint European germoplasm and can be a good alternative to dent hybrid cultivation because white flint grain can lead to higher market prices. In addition, new marketable Reid × Lancaster hybrids have been also identified, as they combined low fumonisin contamination and yields comparable to those exhibited by reference commercial hybrids.

## 5. Materials and Methods

### 5.1. Inbred Selection Program

Pedigree selection was initiated in 2014 using six F_2_ populations derived from crosses among inbreds chosen out of a broad collection of inbred lines based on their low values for FER and fumonisin content [25] (see details in Appendix A). F_2_ populations were obtained from crosses EP31 × EP39 and F575 × EP65 (involving inbreds from the flint European heterotic group), B93 × Oh43 and A670 × H95 (Lancaster group), and A630 × A635 and A654 × A666 (Reid group). Plant selection was performed among F_2_ plants, and family and plant within family selection was carried out during three subsequent self-crossing generations. Selection was performed under the kernel inoculation of the self-crossed ears. Ears were inoculated 7–14 days after self-pollination with a suspension of spores (10^6^ spores/mL) of a strain of *F. verticillioides* deposited in the Culture Collection of the Misión Biológica de Galicia (CSIC-Spain) as MBG-1 [34]. This fungal isolate is an aggressive toxigenic isolate adapted to the local environment. At harvest, FER was estimated according to a 7-point scale (1 = no visible disease symptoms, 2 = 1–3%, 3 = 4–10%, 4 = 11–25%, 5 = 26–50%, 6 = 51–75%, and 7 = 76–100% of kernels exhibiting visual symptoms of infection) formulated by Reid and Zhu [35] and ears with the lowest values for FER (<3) were selected in each self-crossing generation resulting in the development of twenty-nine second-cycle F_6_ inbreds.

### 5.2. Inbred Evaluations

The second-cycle F_6_ inbreds and their parental inbreds were evaluated in 2019 for FER, and the two most promising second-cycle inbreds from each F_2_ together with parental inbreds were analyzed for fumonisin content. In 2021 and 2022, the evaluation focused on selected second-cycle inbreds (Table 1) and their parentals. Split-plot designs with two replicates were used for inbred line evaluation; main plots were assigned to inbred groups, with inbreds derived from the same cross constituting a group. Within each group, sub-plots were assigned to the different inbreds of the group. Each sub-plot consisted of a nine-hill row, with hills spaced 0.18 m apart within the row and 0.8 m between rows. Two kernels were sown in each hill to guarantee a plant density of approximately 70,000 plants ha^−1^ after thinning to one plant per hill. As natural kernel infection by *F. verticillioides* cannot guarantee sufficient and homogeneous inoculum levels, plants were artificially inoculated using a kernel inoculation technique [35,36]. In each row, 7–14 days after the silking date (the date on which more than 50% of plants in the row showed silks), all primary ears (nine) were inoculated with 2 mL of a spore suspension (10^6^ spores per ml) of the fungal isolate of *F. verticillioides* used. Ears from each row were collected two months after inoculation and were individually rated for FER using the seven-point scale previously mentioned. Then, ears were dried at 35 °C for one week and shelled, and a representative kernel sample of approximately 60 g was ground and stored at 4 °C. Kernels were ground through a 0.75 mm screen in a Pulverisette 14 rotor mill (Fritsch GmbH, Oberstein, Germany). Ground samples were analyzed for total fumonisin content (fumonisins B_1_, B_2_, and B_3_) using a commercial ELISA kit (R-Biopharm Rhône Ltd., Glasgow, UK). The recovery rate of the test was approximately 60% with a mean coefficient of variation of approximately 8%; specificities for B_1_, B_2_, and B_3_ were 100%, and approximately 40% and 100%, respectively, and the detection limit was 0.125 mg kg^−1^. Extraction and preparation of samples, as well as test performance, were carried out as described in the kits.

### 5.3. Hybrid Evaluations

Attending to hybrid evaluation, twenty-seven crosses among second-cycle inbred lines and four commercial hybrids were tested in adjacent trials in 2021 and 2022. A complete randomized block design with two replications was used for each hybrid trial. The experimental plot consisted of a single 25-plant row, with the plants being spaced 0.18 m apart. The distance between adjacent rows was 0.8 m. In each row, 7–14 days after the silking date the primary ears of six plants were kernel-inoculated. Two months later, inoculated ears were harvested. The data recorded on each experimental plot were days to shedding pollen (number of days between sowing and the date on which 50% of plants had visible anthers); days to silking (number of days between sowing and the date on which 50% of plants had visible silks), kernel moisture, grain yield expressed in Mg ha^−1^, FER score for each inoculated ear using the visual 7-point scale and fumonisin content of the inoculated ears following the protocol explained above for inbred samples.

### 5.4. Statistical Analyses

An analysis of variance across years was performed for FER and fumonisin content recorded in inbred trials using the Proc Mixed procedure of SAS [37]. Group and inbred (group) were considered fixed effects and year, with replication and all interactions considered random effects. Similarly, an analysis of variance was computed with the same procedure for all data collected in hybrid trials, the hybrid being fixed and replication, and the year and the interaction year × hybrid being random. Least square mean comparisons among hybrids were performed using the Fisher’ protected least significant difference (LSD). In addition, mid-parent values and mid-parent heterosis were estimated for all crosses among the 12 released inbreds (22 hybrids). Mid-parent heterosis was computed as the difference between the hybrid and the mid-parent values divided by the mid-parent value. Pearson correlation coefficients among traits were performed with the procedure Proc CORR of SAS using average inbred and hybrid values.

## Figures and Tables

**Figure 1 toxins-15-00444-f001:**
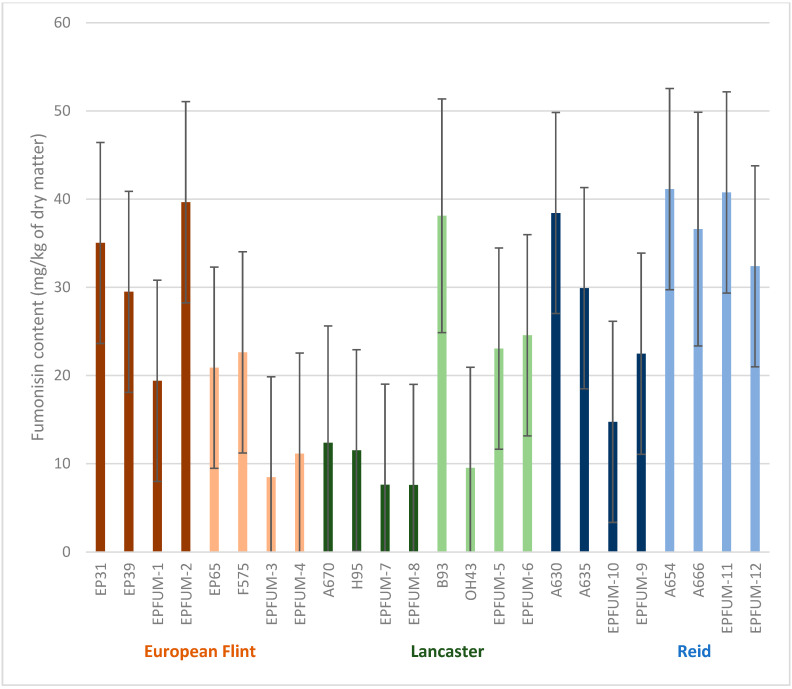
Least square means (±standard error) of the 12 released second-cycle inbreds (EPFUM) and their parents for fumonisin content evaluated across three years. Inbred derived from the same cross and inbreds involved in that cross are in the same color (brown for European flint, green for Lancaster and blue for Reid inbreds).

**Figure 2 toxins-15-00444-f002:**
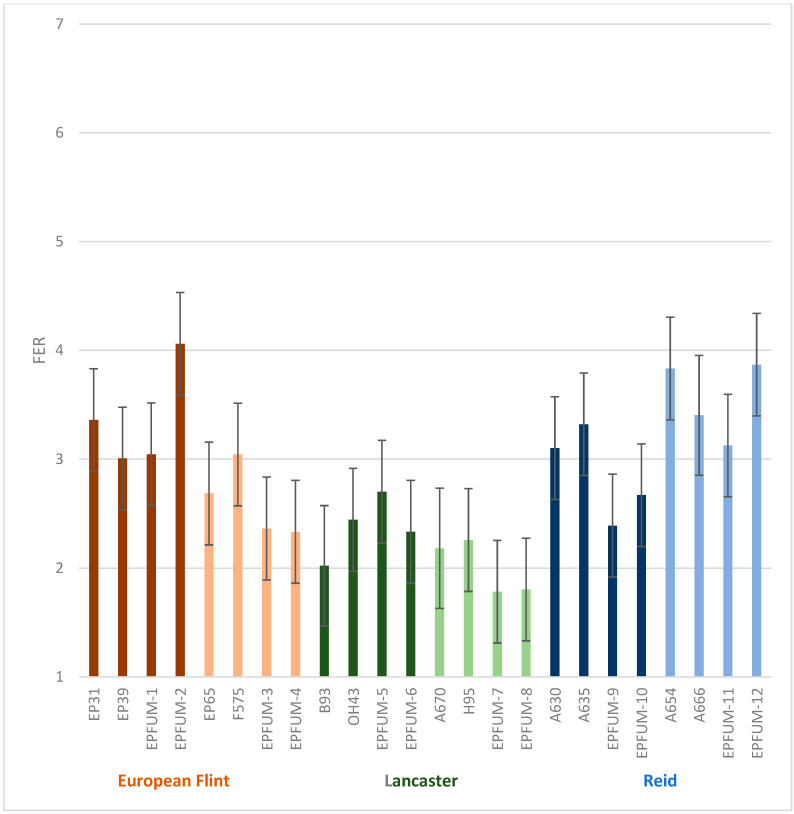
Least square means (±standard error) of the 12 released second-cycle inbreds (EPFUM) and their parents for Fusarium ear rot (FER) evaluated across three years. Inbred derived from the same cross and inbreds involved in that cross are in the same color (brown for European flint, green for Lancaster and blue for Reid inbreds).

**Figure 3 toxins-15-00444-f003:**
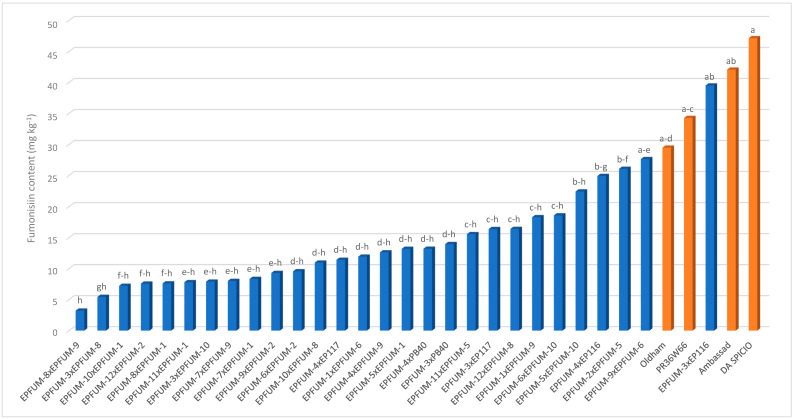
Means of the hybrids for fumonisin content evaluated in two years under inoculation with *Fusarium verticilllioides*. Means with the same letter did not significantly differ at the 0.05 probability level. Commercial hybrids are in orange.

**Figure 4 toxins-15-00444-f004:**
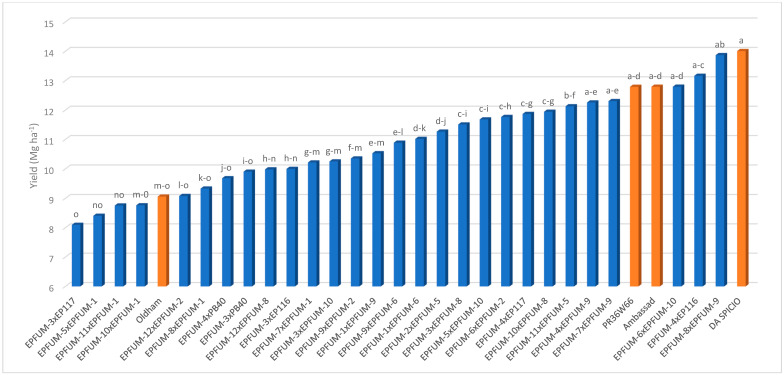
Means of the hybrids for grain yield evaluated in two years under inoculation with *Fusarium verticilllioides*. Means with the same letter did not significantly differ at the 0.05 probability level. Commercial hybrids are in orange.

**Table 1 toxins-15-00444-t001:** Released maize inbred lines (EPFUM) and their parental inbreds evaluated in three years under inoculation with *Fusarium verticillioides*.

Group ^1^	Released Inbred	Parent 1	Parent 2	Heterotic Group	Kernel Color
EP31 × EP39	EPFUM-1	EP31	EP39	European flint	yellow
EP31 × EP39	EPFUM-2	EP31	EP39	European flint	yellow
F575 × EP65	EPFUM-3	F575	EP65	European flint	white
F575 × EP65	EPFUM-4	F575	EP65	European flint	white
B93 × Oh43	EPFUM-5	B93	Oh43	Lancaster	yellow
B93 × Oh43	EPFUM-6	B93	Oh43	Lancaster	yellow
A670 × H95	EPFUM-7	A670	H95	Lancaster	yellow
A670 × H95	EPFUM-8	A670	H95	Lancaster	yellow
A630 × A635	EPFUM-9	A630	A635	Reid	yellow
A630 × A635	EPFUM-10	A630	A635	Reid	yellow
A654 × A666	EPFUM-11	A654	A666	Reid	yellow
A654 × A666	EPFUM-12	A654	A666	Reid	yellow

^1^ Second-cycle inbreds derived from the same cross are clustered in the same group as well as are the inbreds involved in that cross.

**Table 2 toxins-15-00444-t002:** Z- and F-values of the analysis of variance of 28 hybrids involving crosses between the released EPFUM inbred lines and four commercial hybrids evaluated for resistance and agronomical traits in two years under inoculation with *Fusarium verticillioides*.

Source of Variation	Days to Shedding Pollen	Days to Silking	Kernel Moisture	Yield	FER	Fumonisin Content
Random (Z-value)						
Year	0.68	0.69	0.68	0.68	0.18	
Replication (year)	0.95	0.94		0.75	0.88	0.40
Year × Hybrid	1.37	1.88 *	0.50	0.53	2.81 **	
Residual	5.57 **	5.57 **	5.62 **	5.51 **	5.53 **	6.82 **
Fi×ed (F-value)						
Hybrid	18.40 **	16.95 **	2.56 **	6.56 **	1.44	2.68 **

*, ** Z- or F-values significant at 0.05 and 0.01 probability levels, respectively.

**Table 3 toxins-15-00444-t003:** Least square means of 28 maize hybrids involving crosses of EPFUM inbreds and four commercial hybrids for days to shedding pollen, Fusarium ear rot (FER) and kernel moisture evaluated at harvest for two years.

Hybrid	FER (1–7)	Kernel Moisture (%)	Days to Shedding Pollen
Ambassad	2.9 a	17.1 e–g	73.2 bc
Da SPICIO	3.5 a	17.3 e–g	72.5 c–e
PR36W66	3.2 a	18.0 d–g	75.8 a
Oldham	3.3 a	15.9 fg	63.3 l
EPFUM-1 × EPFUM-6	2.5 a	23.0 ab	69.3 g–i
EPFUM-1 × EPFUM-9	2.7 a	19.8 c–e	70.8 d–h
EPFUM-2 × EPFUM-5	2.2 a	19.9 c–e	65.8 kj
EPFUM-3 × EP116	3.4 a	19.1 c–f	71.5 c–f
EPFUM-3 × EP117	2.9 a	19.2 c–f	69.3 g–i
EPFUM-3 × EPFUM-10	2.4 a	18.1 d–g	72.0 c–e
EPFUM-3 × EPFUM-8	1.8 a	19.0 d–f	72.8 cd
EPFUM-3 × PB40	3.0 a	19.1 c–f	70.8 d–h
EPFUM-4 × EP116	3.2 a	19.1 d–f	71.0 d–g
EPFUM-4 × EP117	2.9 a	19.4 c–e	69.5 f–i
EPFUM-4 × EPFUM-9	2.6 a	17.8 d–g	71.8 c–e
EPFUM-4 × PB40	2.8 a	19.6 c–e	72.0 c–e
EPFUM-5 × EPFUM-1	2.3 a	19.3 c–e	64.5 kl
EPFUM-5 × EPFUM-10	3.4 a	19.6 c–e	70.5 e–h
EPFUM-6 × EPFUM-10	3.4 a	19.1 d–f	71.3 c–g
EPFUM-6 × EPFUM-2	2.1 a	23.8 a	68.8 ih
EPFUM-7 × EPFUM-1	2.0 a	17.3 e–g	67.8 ij
EPFUM-7 × EPFUM-9	1.8 a	17.3 e–g	75.3 ab
EPFUM-8 × EPFUM-1	1.7 a	20.5 b–d	68.0 i
EPFUM-8 × EPFUM-9	2.4 a	19.4 c–e	71.8 c–e
EPFUM-9 × EPFUM-2	2.5 a	15.4 f	68.3 i
EPFUM-9 × EPFUM-6	2.5 a	22.2 a–c	72.5 c–e
EPFUM-10 × EPFUM-1	2.0 a	19.7 c–e	65.3 kl
EPFUM-10 × EPFUM-8	3.1 a	18.7 d–f	70.5 e–h
EPFUM-11 × EPFUM-1	2.3 a	19.5 c–e	64.8 kl
EPFUM-11 × EPFUM-5	2.9 a	19.8 c–e	67.8 ij
EPFUM-12 × EPFUM-2	2.3 a	19.9 c–e	64.3 kl
EPFUM-12 × EPFUM-8	3.3 a	19.3 c–e	68.3 i

Means in the same column followed by the same letter did not significantly differ at the 0.05 probability level.

**Table 4 toxins-15-00444-t004:** Mid-parent and heterosis estimates for FER and fumonisin content computed with data collected on the 22 hybrids involving crosses between EPFUM inbreds.

Hybrid	Mid-Parent ^1^		Mid-Parent Heterosis	
	FER	Fumonisin	FER	Fumonisin
EPFUM-1 × EPFUM-6	2.70	22.0	−0.0859	−0.4575
EPFUM-1 × EPFUM-9	2.72	21.0	−0.0035	−0.1263
EPFUM-2 × EPFUM-5	3.38	31.3	−0.3592	−0.1684
EPFUM-3 × EPFUM-8	2.08	8.0	−0.1203	−0.3218
EPFUM-3 × EPFUM-10	2.52	11.6	−0.0397	−0.3192
EPFUM-4 × EPFUM-9	2.36	16.8	0.1113	−0.2488
EPFUM-5 × EPFUM-1	2.87	21.2	−0.2169	−0.3788
EPFUM-5 × EPFUM-10	2.69	18.9	0.2567	0.1870
EPFUM-6 × EPFUM-2	3.20	32.1	−0.3485	−0.7021
EPFUM-6 × EPFUM-10	2.50	19.7	0.3490	−0.0550
EPFUM-7 × EPFUM-1	2.41	13.5	−0.1716	−0.3829
EPFUM-7 × EPFUM-9	2.09	15.0	−0.1416	−0.4670
EPFUM-8 × EPFUM-1	2.42	13.5	−0.3125	−0.4359
EPFUM-8 × EPFUM-9	2.10	15.0	0.1322	−0.7858
EPFUM-9 × EPFUM-2	3.23	31.1	−0.2380	−0.7006
EPFUM-9 × EPFUM-6	2.36	23.5	0.0582	0.1750
EPFUM-10 × EPFUM-1	2.86	17.1	−0.3000	−0.5773
EPFUM-10 × EPFUM-8	2.24	11.2	0.3635	−0.0184
EPFUM-11 × EPFUM-1	3.09	30.1	−0.2707	−0.7407
EPFUM-11 × EPFUM-5	2.91	31.9	−0.0133	−0.5124
EPFUM-12 × EPFUM-2	3.96	36.0	−0.4220	−0.7896
EPFUM-12 × EPFUM-8	2.84	20.0	0.1487	−0.1797

^1^ The mid-parent estimate was calculated as the average performance across years of both inbred parents of the hybrid. The mid-parent heterosis estimate was calculated as the difference between the hybrid mean and the mid-parent value divided by the mid-parent value.

**Table 5 toxins-15-00444-t005:** Pearson correlation coefficients among traits recorded in the 32 hybrids across two years.

	Days to Silking	Kernel Moisture	Yield	FER	Fumonisin Content
Days to shedding pollen	0.95 **	−0.11	0.64 **	0.22	0.28
Days to silking		−0.06	0.51 **	0.23	0.23
Kernel moisture			−0.10	−0.24	−0.21
Yield				0.29	0.32
FER					0.66 **

**** significant at 0.01 probability level.

**Table 6 toxins-15-00444-t006:** Pearson correlation coefficients between hybrid performance and mid-parent and heterosis estimates for Fusarium ear rot (FER) and fumonisin content computed with data collected on the 22 hybrids involving crosses between EPFUM inbreds.

	Mid-Parent ^1^		Heterosis	
	*r* ^2^	*p*-Value	*r* ^2^	*p*-Value
FER	0.02	0.94	0.80	<0.0001
Fumonisin content	0.29	0.20	0.76	<0.0001

^1^ The mid-parent estimate was calculated as the average performance across years of both inbred parents of the hybrid. The mid-parent heterosis estimate was calculated as the difference between the hybrid mean and the mid-parent value divided by the mid-parent value.

## Data Availability

Data will be made available upon request.

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
