# Peer review of "Inbred Selection for Increased Resistance to Kernel Contamination with Fumonisins"

_toxins, 2023, doi:10.3390/toxins15070444_

Round 1
Reviewer 1 Report
The manuscript is well written with only minor revisions needed.
This research, like others addressing this topic show gains can be made for reducing Fusarium ear rot and the production of fumonisins through line recycling. Some outstanding sources can be identified, but more frequently gains are incremental and require continued phenotypic selection to increase these often-small gains through each cycle of selection.
Suggested modifications
Line 5 Change to kernels are
Line 16 Change to improve
Line 28 Change to kernels are
Line 29 Change to Fusarium verticillioides, where accumulation
Line 51 Change to per se and in testcrosses
Line 56 Change to address the farmer's needs
Line 104 Change to released EPFUM lines
Line 108-109 Change to kernel moisture evaluated at harvest for two years.
Page 10, Discussion, Line 3 Change to seem to be a consequence of
Page 10, Discussion, Last line Change to Hybrid EPFUM-4 x EP116 can be an alternative higher value specialty crop and add genetic variability to the current narrow based commercial dent hybrids being grown [32].
Page 11, Materials and Methods, Line 8 Should this be 7-14 days to be consistent with line 26?
Page 13, References, reference 10 should be "... and corresponding"
Minor grammatical errors have been noted above.
Author Response
All suggested modifications has been included in the revised version
Reviewer 2 Report
The manuscript titled “Inbred Selection for Increased Resistance to Kernel Contamination with Fumonisins” focuses on a breeding program aimed at obtaining inbred lines with resistance to fumonisin contamination in maize kernels. The manuscript presents the results of evaluating the released inbreds and hybrids derived from them, with a particular emphasis on their resistance to Fusarium Ear Rot (FER) and fumonisin contamination.
While the study provides some valuable insights into the resistance levels of the tested inbreds and hybrids, there are several significant drawbacks that need to be addressed:
1. The objectives of the study are not clearly defined in the abstract. The authors mention assessing the resistance levels of released inbreds and examining the competitiveness of hybrids derived from these inbreds, but the specific criteria for determining competitiveness are not stated. Clarity in the objectives is crucial for understanding the scope and relevance of the study.
2. The manuscript lacks detailed information about the methodology used for evaluating resistance to FER and fumonisin contamination. Important details such as the sample size, experimental design, and statistical analyses are not provided. Without this information, it is difficult to assess the robustness and reliability of the results.
3. The study mentions that most of the differences observed between the second-cycle inbreds and their parental inbreds in terms of resistance to fumonisin contamination were not statistically significant. However, the manuscript does not discuss the potential implications of these nonsignificant differences, which is necessary for drawing meaningful conclusions.
4. The manuscript briefly mentions previous studies on breeding for resistance to FER and fumonisin accumulation but does not provide a comprehensive discussion or comparison of the current findings with the existing literature. This limits the ability to assess the novelty and contribution of the study.
5. The manuscript lacks a thorough discussion of the limitations of the study. For example, the authors acknowledge the reliance on FER as an indirect selection trait but do not discuss the potential limitations and drawbacks of using FER as a proxy for fumonisin contamination. Additionally, the manuscript does not address potential limitations related to the specific genotypes evaluated or the environmental conditions under which the study was conducted.
6. The conclusion section of the manuscript is brief and does not provide a comprehensive summary of the findings. It does not discuss the implications of the results for future breeding efforts or suggest avenues for further research.
7. The manuscript contains several grammatical errors, formatting inconsistencies, and unclear sentences, which detract from the overall readability and professionalism of the work. A thorough proofreading and editing process should have been conducted to ensure clarity and coherence.
8. The references generally follow the format of the journal; however, these should be crosschecked.
The manuscript contains several grammatical errors, formatting inconsistencies, and unclear sentences, which detract from the overall readability and professionalism of the work. A thorough proofreading and editing process should have been conducted to ensure clarity and coherence.
Author Response
While the study provides some valuable insights into the resistance levels of the tested inbreds and hybrids, there are several significant drawbacks that need to be addressed:
- The objectives of the study are not clearly defined in the abstract. The authors mention assessing the resistance levels of released inbreds and examining the competitiveness of hybrids derived from these inbreds, but the specific criteria for determining competitiveness are not stated. Clarity in the objectives is crucial for understanding the scope and relevance of the study.
Objectives have been changed to: “The objectives were (i) to assess if released inbreds by that breeding program were significantly more resistant than their parental inbreds and (ii) to examine if hybrids derived from EPFUM inbreds could be competitive based on grain yield and resistance to FER and fumonisin contamination”
- The manuscript lacks detailed information about the methodology used for evaluating resistance to FER and fumonisin contamination. Important details such as the sample size, experimental design, and statistical analyses are not provided. Without this information, it is difficult to assess the robustness and reliability of the results.
Methodology is at the end of the paper following the rules of Toxins. Details for inbred trial design are in lines 115-118 and for hybrid trials in lines 137-139. The number of ears sampled in the inbred trials was 9 per plot and 6 per plot in the hybrid trials. Statistical analyses are explained in lines 149-160.
- The study mentions that most of the differences observed between the second-cycle inbreds and their parental inbreds in terms of resistance to fumonisin contamination were not statistically significant. However, the manuscript does not discuss the potential implications of these nonsignificant differences, which is necessary for drawing meaningful conclusions.
We have clarified in the text: ”In the current study, second-cycle inbreds obtained through pedigree selection program did not significantly improve the resistance levels to fumonisin contamination of their parental inbreds; we have to take into account those parents were already the most resistant to fumonisin contamination among a diverse collection of inbred lines with more than 200 accessions [24]”
- The manuscript briefly mentions previous studies on breeding for resistance to FER and fumonisin accumulation but does not provide a comprehensive discussion or comparison of the current findings with the existing literature. This limits the ability to assess the novelty and contribution of the study.
We have compared our results with all those obtained specifically by applying inbred selection for FER. We could also mention results obtained by applying recurrent selection, although this has not been the focus of the present paper.
- The manuscript lacks a thorough discussion of the limitations of the study. For example, the authors acknowledge the reliance on FER as an indirect selection trait but do not discuss the potential limitations and drawbacks of using FER as a proxy for fumonisin contamination. Additionally, the manuscript does not address potential limitations related to the specific genotypes evaluated or the environmental conditions under which the study was conducted.
We think the most important limitation of the breeding selection procedure used was that selection was based on single plant performance and low heritability estimates on an individual plant basis were reported for FER. We do not think that performing indirect selection using FER can limit gains for resistance to fumonisins contamination because genetic correlation coefficients between both traits are moderate to high. This is already discussed in the manuscript (Page 10 line 23-35). Some limitations attending to the parents used have been also added.
- The conclusion section of the manuscript is brief and does not provide a comprehensive summary of the findings. It does not discuss the implications of the results for future breeding efforts or suggest avenues for further research.
We have included a more conclusive statement attending to future breeding efforts Page 11. Lines 80-85: Second-cycle inbred lines from the three heterotic groups tested showed a non-significant improvement in resistance to FER and fumonisin contamination compared to their parental inbreds, probably due to the low inheritance of maize resistance to FER. Therefore, breeding schemes using replicated-family evaluations for FER or genomic selection would be new approaches to be explored.”
- The manuscript contains several grammatical errors, formatting inconsistencies, and unclear sentences, which detract from the overall readability and professionalism of the work. A thorough proofreading and editing process should have been conducted to ensure clarity and coherence.
Proofreading and editing have been performed
- The references generally follow the format of the journal; however, these should be crosschecked. References have been checked
Reviewer 3 Report
This research is a long-term project and the result is meaningful for breeding. The main problems of this paper are listed as follows:
(1) It is strange that there is no data on protein content or something else, since grain quality is pretty important trait for breeding.
(2) It is necessary to describe the 12 parental inbreds in the introduction or methods part, such as yield, FER resistance, quality and the key agronomic traits.
(3) Can the four commercial hybrids be representative enough?
(4) Please explain why the cross was made as Tables 1 and 3?
(5) Is there any QTL on FER resistance of the parental inbreds?
(6) The standard errors in figures 1 and 2 are so big. How about the comparsion in same year? significant or still not? Does it means the project is insuccessful?
(7) There is no negative correlation between yield and FER, and between yield and Fumonisin content? what does it mean?
(8) The first paragraph in materials and methods should be separated. Please give brief title for different contents of this part, to facilitate understanding.
Author Response
This research is a long-term project and the result is meaningful for breeding. The main problems of this paper are listed as follows:
(1) It is strange that there is no data on protein content or something else, since grain quality is pretty important trait for breeding.
Our main objective was to characterize the genotype's risk of fumonisin contamination. Further characterization attending added values will be explored in subsequent studies.
(2) It is necessary to describe the 12 parental inbreds in the introduction or methods part, such as yield, FER resistance, quality and the key agronomic traits.
A supplementary table has been included.
(3) Can the four commercial hybrids be representative enough?
We consider hybrids representative because they represent different FAO cycles and have been extensively grown in the target region.
(4) Please explain why the cross was made as Tables 1 and 3?
I do not understand this question
(5) Is there any QTL on FER resistance of the parental inbreds?
We have done QTL studies for FER and fumonisins content but none of the parental inbreds were used to build the mapping populations used.
(6) The standard errors in figures 1 and 2 are so big. How about the comparsion in same year? significant or still not? Does it means the project is insuccessful?
As the inbred x year interaction was not significant, mean comparisons were done across years. Although differences among means were not significant, some of the released inbred tended to improve parental characteristics and the hybrids among the released inbreds performed better for Fusarium ear rot and fumonisin content than checks. Therefore, although genetic gains were not significant, some hybrids seemed highly valuable
(7) There is no negative correlation between yield and FER, and between yield and Fumonisin content? what does it mean?
That means that we do not expect an unfavorable effect on yield when performing selection for FER, meanwhile yield reduction is common when performing selection for resistance to pests.
(8) The first paragraph in materials and methods should be separated. Please give brief title for different contents of this part, to facilitate understanding.
Separation of contents was done.
To finish, we would like to thank reviewers for their value comments. We hope that you find our revision and responses to be acceptable.